# Fabrication and Characterization of W/O/W Emulgels by *Sipunculus nudus* Salt-Soluble Proteins: Co-Encapsulation of Vitamin C and β-Carotene

**DOI:** 10.3390/foods11182720

**Published:** 2022-09-06

**Authors:** Yaping Dai, Xuli Lu, Ruyi Li, Yupo Cao, Wei Zhou, Jihua Li, Baodong Zheng

**Affiliations:** 1Key Laboratory of Tropical Crop Products Processing of Ministry of Agriculture and Rural Affairs, Agricultural Products Processing Research Institute, Chinese Academy of Tropical Agricultural Sciences, Zhanjiang 524001, China; 2Hainan Key Laboratory of Storage & Processing of Fruits and Vegetables, Zhanjiang 524001, China; 3College of Food Science, Fujian Agriculture and Forestry University, Fuzhou 350002, China

**Keywords:** W/O/W emulgels, *Sipunculus nudus*, salt-soluble proteins, vitamin C, β-carotene, co-encapsulation

## Abstract

W/O/W emulsions can be used to encapsulate both hydrophobic and hydrophilic bioactive as nutritional products. However, studies on protein stabilized gel-like W/O/W emulsions have rarely been reported, compared to the liquid state multiple emulsions. The purpose of this study was to investigate the effect of different oil–water ratios on the stability of W/O/W emulgels fabricated with salt-soluble proteins (SSPs) of *Sipunculus nudus*. The physical stability, structural characteristics, rheological properties, and encapsulation stability of vitamin C and β-carotene of double emulgels were investigated. The addition of W/O primary emulsion was determined to be 10% after the characterization of the morphology of double emulsion. The results of microstructure and rheological properties showed that the stability of W/O/W emulgels increased with the increasing concentration of SSPs. Additionally, the encapsulation efficiency of vitamin C and β-carotene were more than 87%, and 99%, respectively, and still could maintain around 50% retention of the antioxidant capacity after storage for 28 days at 4 °C. The aforementioned findings demonstrate that stable W/O/W emulgels are a viable option for active ingredients with an improvement in shelf stability and protection of functional activity.

## 1. Introduction

Double emulsions are a type of multiple emulsion in which oil-in-water (O/W) and water-in-oil (W/O) co-exist, offering more advantages than conventional water-in-oil (W/O) emulsions in the delivery system of bioactives [1,2,3]. Typically, they are split into W/O/W and O/W/O multiple emulsions [4]. Double emulsions are gaining interest in the food and drug delivery industry for encapsulating salt (salted creams), spices (condiments and sauces), enzymes, vitamins, carotenoids, and omega-3 oils with improved stability [5,6,7,8]. However, their actual applications have relatively few examples because of their tendencies of coalescence, flocculation, and aggregation [9]. Therefore, there are several important strategies to keep the long-term storage stability of W/O/W emulsions. The crux of these strategies is balancing the osmotic pressure of the internal and external aqueous phases to provide good aggregation stability or reduce droplet mobility by increasing the fraction of the middle oil phases to prevent droplet coalescence, expansion, or collapse [10,11,12]. Currently, an interesting trend in W/O/W emulsions research is the utilization of natural emulsifiers and different oil–water ratios for the gel-like emulsion systems fabrication to enhance their stability. Jiang et al. (2020) [13] found that zein nanoparticles as hydrophilic emulsifiers could improve the stability of forming W/O/W emulsions under the high oil–water ratio (W_1_:O:W_2_ = 0.3:3:0.7). Tang et al. (2021) [9] used the fabricated sugar beet pectin-bovine serum albumin nanoparticles (SBNPs) as a particle stabilizer to develop W/O/W emulsion gels, which provided insights for improving the storage stability of bioactive substances. Consequently, the addition of natural proteins as emulsifiers to improve the stability of double emulgels has gradually attracted the attention of researchers.

*Sipunculus nudus* (*S. nudus*) belongs to marine worms, which are rich in numerous active ingredients such as protein, polysaccharides, and fatty acids [14,15]. Previously, it has been reported for its immunomodulatory, antioxidant, and anti-fatigue biological activity, and so on [16]. However, there are few studies about the functional properties of the *S. nudus* protein. In our previous study, the crude protein of *S. nudus* as the sole emulsifier could successfully develop stable high internal phase emulsion (HIPE) [15]. However, such a soluble surfactant requires high concentrations to maintain stable gel-like emulsions. Interestingly, studies have found that salt-soluble proteins (SSPs) can dissolve and swell (protein hydration) and create a soluble gel during the chopping process. These SSPs can adsorb onto fat particles and form interfacial protein films to improve emulsification stability [17]. The interfacial protein membranes can emulsify and fix the fat particles, and a cohesive protein gel matrix can bind and limit the flow of fat and water [18]. Therefore, in this study, SSPs of *S. nudus* were used to prepare the W/O/W emulgels and enhance the stability of encapsulation of nutraceuticals.

Vitamin C, a water-solution substance, is obtained from fruits, vegetables, and their products. In addition, vitamin C is used in the food and cosmetic industry as an antioxidant and nutrient [19]. Simultaneously, β-carotene is a model hydrophobic nutraceutical for the properties of its potential as both a nutraceutical and natural food colorant [20]. Nevertheless, vitamin C and β-carotene have poor chemical stability under thermal conditions, light conditions, and storage processes in the food industry, which greatly limits their applications and reduced their health benefits.

Herein, the main objective of our study was to prepare the stable W/O/W emulgels using *S. nudus* protein as an emulsifier of the outer aqueous phase at different oil–water ratios. The microstructure and properties of double emulsions were analyzed by particle size, microscopy, and dynamic rheology property measurements. In addition, to further demonstrate that the emulsion system could be used as a carrier for active compounds, the encapsulation rates of vitamin C and β-carotene by W/O/W emulgels were also evaluated as well as the antioxidant activity under different storage conditions. The results of this study might provide a potential application for producing edible and more stable delivery systems of double emulsions using protein-based emulsifiers.

## 2. Materials and Methods

### 2.1. Materials

Fresh *S. nudus* were supplied by Dongfeng market in Zhanjiang, Guangdong, China. Soybean oil (purity >98%, Arawan) was purchased from Wal-Mart (Zhanjiang, China). Polyglycerol polyricinoleate (PGPR, >75%) was acquired from Shanghai Yuanye Industrial Co., Ltd. (Shanghai, China). Fluorescein isothiocyanate (FITC, 95%, HPLC) and Nile red (99%) were purchased from Sigma-Aldrich Co., Ltd. (St. Louis, MO, USA). β-Carotene (>96.0%) and Vitamin C (>99.7%) were obtained from Aladdin Industrial Co., Ltd. (Shanghai, China). All other reagents were of analytical grade.

### 2.2. Extraction and Analysis of Salt Soluble Protein (SSPs) from S. nudus

SSP extraction was carried out with slight modification according to the method of Han et al. (2020) [21]. Firstly, the body wall of fresh *S. nudus* was ground using a meat grinder (SD-JR39, Foshan Shunde Sandi Electric Appliance Manufacturing Co., Ltd., Foshan, Zhanjiang, China), and mixed with distilled water (Milli-Q grade, 15.0 MΩ cm) at 1:2 (*w*/*w*) and left at 4 °C for 12 h. The mixture was washed twice with distilled water and then centrifuged at 9569× *g* for 15 min at 4 °C (Sigma/3-30K, Sigma, Osterode am, Harz, Germany). Secondly, the precipitation was collected and dissolved in 0.8 mol/L NaCl solution at 1:4 (*w*/*v*), and stirred overnight at 4 °C. Then, the solution was centrifuged at 9569× *g* for 15 min at 4 °C. Finally, the supernatants were dialyzed with the 1.0 kDa dialysis bag in distilled water for 72 h. The dialysate was lyophilized (Alpha 1–4 LDplus, Chirst, Germany) to produce SSP powders. The purity of SSPs was 92.5% by Dumas Nitrogen Analyzer (D50, Haineng, Jinan, China). The mean particle size of SSPs was 995.2 ± 50.5 nm.

### 2.3. Preparation of the Emulsion

W_1_/O/W_2_ emulgels were prepared by a two-step process. The primary emulsion (W_1_/O) was first prepared and then the primary emulsion and soybean oil were added to the outer aqueous phase (W_2_) to prepare double emulgels [13].

#### 2.3.1. Preparation of the Primary Emulsion (W_1_/O)

The primary emulsion was prepared by adding the phosphoric acid buffer solution (PB, W_1_, 0.1 mM, pH 7.4) into soybean oil containing 3.0% (*w*/*w*) PGPR. The W_1_ (4 mL) was mixed with soybean oil (16 mL) under magnetic stirring at 1000 rpm for 1 min (RT10, IKA, Staufen, Germany), followed by homogenizing at 12,000 rpm for 3 min using a high-speed homogenizer (Ultraturrax T18, IKA, Staufen, Germany). The ratio of oil–water was kept as 2:8.

#### 2.3.2. Type Analysis of Primary Emulsion

The type of primary emulsion was determined by dropping the prepared primary emulsion (Section 2.3.1) into oil or water for observation. An emulsion was considered an oil-in-water (O/W) emulsion if the droplets dissolve in water and accumulate in the oil, and if it was insoluble in water, it was considered a water-in-oil (W/O) emulsion [22].

#### 2.3.3. Preparation of W_1_/O/W_2_ Emulgels

For W_1_/O/W_2_ emulgels, briefly, the SSPs with different concentrations (0.5, 1.0, 1.5, 2.0, and 2.5 wt%) were dissolved in the buffer solution (PB, 0.1 mM, pH 7.4) as a hydrophilic emulsifier (W_2_). The double emulgel (20 g) was prepared by adding the different levels of W_1_/O primary emulsion (5, 10, 15, and 20%, *w*/*w*), soybean oil, and 25% (*w*/*w*) of W_2_ (1.0% SSPs) to a 50 mL beaker, and then homogenized the dispersions by homogenizer at 10,000 rpm for 3 min. The sodium azide (0.01 wt%) was added to the emulsions as an antimicrobial preservative.

### 2.4. Determination of Particle Size and ζ-Potential

The particle size of double emulgels was measured according to previously reported methods with some modifications [20]. The particle size distribution and droplet size (d_3,2_) of the droplets in W/O/W emulgels were determined using laser diffraction (Mastersizer 2000, Malvern Instruments 164 Ltd., Malvern, Worcestershire, UK). The ζ-potential was calculated using a Malvern Zetasizer Nano (ZSU5800, Malvern Instruments, UK). Specifically, the double emulsions were diluted with buffer solutions of the same pH (7.4) to avoid multiple scattering effects. The buffer solutions and the soybean oil mixtures’ respective refractive indices were determined to be 1.330 and 1.474. All measurements were constant in triplicate at 25 °C.

### 2.5. Optical Microscopy and Confocal Laser Scanning Microscopy (CLSM)

The optical micrographs of W/O/W emulgels were observed using an inverted microscope with 20× and 63× objective lens (Leica DMI6000 B, Leica, Heidelberg, Germany). A confocal laser scanning microscope (FV 3000, OLYMPUS, Tokyo, Japan) was used to obtain the fluorescence images of the multiple emulsions. Before observation, the oil phase and the outer aqueous phase were stained with Nile Red and FITC, respectively. The samples were diluted 10 times with distilled water, then a 10 μL drop was placed on a glass slide and covered with a cover slip for observation.

### 2.6. Rheological Properties

The rheological properties were measured using a dynamic rheometer (MARS III, TA Instruments, New Castle, DE, USA) with a parallel plate geometry (steel parallel plate: P 35 mm, gap: 0.5 mm) according to the method of Cao et al. (2021) [23] with some modifications. The frequency sweep test was enforced to analyze the dynamic stress–strain relationships with angular frequency from 0.1 to 100 rad/s at a fixed shear strain of 1.0% (with the LVR). The shear sweep test was conducted from 0.1 to 100 s^−1^ to study the change in the shear rate on the apparent viscosity (η) of emulsion. All samples were measured at 25 °C.

### 2.7. Storage Stability

The fresh W/O/W emulgels were prepared and stored at 4 °C for 0, 10, and 30 days for stability measurements. The visual appearance and optical micrographs were obtained to record the phenomenon of cream or sedimentation of emulsions during storage.

### 2.8. Encapsulation of Vitamin C and β-Carotene

To investigate the encapsulation stability of multiple emulsions, encapsulation yield and antioxidant activity test of vitamin C (20 mg/mL) in the inner aqueous phase (W_1_) and β-carotene (1 mg/mL) in the oil phase of the W_1_/O/W_2_ emulgels were conducted as follows.

#### 2.8.1. Encapsulation Efficiency (EE)

The encapsulation efficiency of the double emulgels was determined based on Hang et al. (2019) [24] with slight modification. Briefly, 10 mL of the fresh double emulsion formed was centrifuged at 2392× *g* for 10 min at 4 °C. The separated phase (the aqueous phase and oil phase) was collected, and the encapsulation efficiency was determined. The β-carotene content was identified according to the previous method [25]. The samples were diluted with *n*-hexane/absolute ethanol (1:2, *v*/*v*), and then measured at 450 nm by an ultraviolet–visible spectrophotometer (U-T6A, Yipu Instrument Manufacturing Co., Ltd., Shanghai, China). A linear calibration curve was established using different β-carotene concentrations (0.5–5.0 µg/mL, y = 0.1948x + 0.0038, R^2^ = 0.995). Similarly, the vitamin C content was determined in high-performance liquid chromatography (HPLC, Agilent 1260, Kyoto, Japan) with a C18 analytical chromatography column (250 mm × 4.6 mm, 5 μm; 118 × 20243; Zhongpu science Inc., Fujian, China) as Wang et al. (2019) [19] described, with some slight modifications. Vitamin C was dissolved by dimethyl sulfoxide (DMSO) and then filtered through a 0.22 µm nylon filter prior to analysis. A gradient of the mobile phase composed of methanol (solvent A) and 0.01% (*v*/*v*) phosphoric acid solution (solvent B) was used according to the following program: 0–5 min, 5% A; 5–6 min, 5 to 15% A; 6–8 min, 15% to 35% A; 8–13 min, 35% to 5% A; 13–20 min, 5% A. The measurement was carried out with a flow rate of 0.1 mL/min. The eluate was detected using a model UV detector, set at 245 nm. The injection volume was 10 μL and chromatographed was performed at 25 ± 1 °C. The calibration curve was performed using the standard vitamin C solution (10–600 µg/mL, y = 52359x + 7 × 10^6^, R^2^ = 0.997). The EE of vitamin C and β-carotene could be expressed as Equations (1) and (2):(1)$$\mathrm{EE}\;\mathrm{vitamin}\;C\;\left(\%\right)=\frac{\left(N_{v}-{\;N}_{\mathrm{si}}\right)}{N_{v}}\times 100\%$$
(2)$$\mathrm{EE}\;\beta -\mathrm{carotene}\;\left(\%\right)=\frac{\left(N_{c}-{\;N}_{\mathrm{so}}\right)}{N_{c}}\times 100\%$$
where N_v_ and N_c_ were the amounts of vitamin C and β-carotene added into the emulsion, respectively. N_si_ and N_so_ were the amount of vitamin C and β-carotene in the separated phase, respectively.

#### 2.8.2. Determination of Antioxidant Activity

Antioxidant activity evaluation of vitamin C and β-carotene in W/O/W emulgels was carried out by measuring DPPH· scavenging activity and ABTS+ radical scavenging, based on the previously described method [26]. Encapsulated vitamin C and β-carotene were extracted from double emulgels. The emulsions were mixed and stirred with *n*-hexane and absolute ethanol (1:2, *v*/*v*). The mixtures were centrifuged at 9569× g for 15 min, and the supernatant was collected for the antioxidant activity test.

In the ABTS+ radical scavenging experiment, a stock solution of 0.0384 g of ABTS+ and 0.0065 g of potassium persulfate in deionized water (10 mL) was obtained and incubated for 12–16 h in the dark, followed by diluting to 0.700 (±0.001) with ethanol using a UV–visible spectrometer (UV-8000, Shanghai Yuanxi Instrument Co., Ltd., Shanghai, China). Then, the sample was added to the mixture of ABTS+, and the absorbance was measured at 734 nm after 30 min. The standard curve was determined with Trolox as standard (y = 0.1063x + 0.0008, R^2^ = 0.990). The retention of antioxidant activity (RA, %) of the samples against the ABTS+ free radical was obtained using the following Equation (3):(3)$$\mathrm{RA}=\frac{G_{t}}{G_{0}}\times 100\%$$
where G_0_ and G_t_ (μg Trolox/g sample) are the radical scavenging activity of initial emulsions and of the measured emulsions, respectively.

For the DPPH· scavenging activity test, samples were incubated with 4 mg/100 mL of methanolic solutions of DPPH· in the dark for 30 min, and absorbance was recorded at 517 nm. The DPPH· scavenging activity was also assessed with Trolox as standard (y = 0.0846x + 0.0137, R^2^ = 0.995). The retention of antioxidant activity (RA, %) of the samples against the DPPH· free radical was calculated according to Equation (3).

### 2.9. Statistical Analysis

All the statistical analysis was conducted in triplicate. The data were analyzed using SPSS 16.0 (2010, IBM, Armonk, NY, USA) by ANOVA (*p* < 0.05), and the means values were separated by Duncan’s test.

## 3. Results and Discussion

### 3.1. Characterization of the Primary Emulsion

Figure 1a showed the visual appearance of primary emulsions in oil or water, which displayed that it was dispersed in oil and aggregated in the water phase. Therefore, the primary emulsion was identified as a water-in-oil (W/O) emulsion.

In this part, to explore the possibilities of W/O/W emulgels stabilized by SSPs at different oil–water ratios, the addition of W/O primary emulsion was gradually increased from 5% to 20% while keeping the concentration of SSPs. All fresh double emulsions exhibited gel-like behavior, and the droplet size gradually increased with the increasing addition of primary emulsions from optical micrographs (Figure 1b,c). The percentage of PGPR and internal water phase of W/O/W emulsion gel system increased with the increase addition of primary emulsion. Interestingly, previous studies have reported that the droplet size of the W/O emulsions reduced with increasing PGPR due to its lipophilic emulsification characteristics [27,28]. However, the droplet size of W/O/W emulgels increased from 11.23 to 13.07 μm with increasing primary emulsions from 5% to 20% (Figure 1d). This suggested that the primary emulsion might have influenced the tendency for the double emulsion to aggregate. Here, we proposed two possible reasons for the above phenomenon: (1) Competitive absorption of the excessive PGPR and protein (W_2_) at the external interface during the formation procedure of the double emulsion, which weakened the absorption efficiency of protein. (2) The osmotic pressure difference between the inner and outer aqueous phase was affected by the increase in the inner aqueous phase, which resulted in the diffusion of the inner aqueous phase to the outer aqueous phase [29]. Additionally, there was little change (non-significant) (*p* > 0.05) in the ζ-potential (Figure 1e) values in all double emulsions. It suggested that the interfacial composition of fresh emulsion with different W/O primary emulsion additions was relatively constant [30]. As shown from confocal laser scanning microscopy (CLSM) in Figure 2, it was obvious that the inner aqueous phase, the oil phase, and the outer aqueous phase of double emulgels were not dyed (bleak), Nile red dyed (red), and FITC dyed (green), respectively. Thus, the double emulgel was a type of water-in-oil-in-water (W/O/W) emulsion.

Rheological properties could provide some valuable information for the application of emulsions [25]. Frequency sweep curves and apparent viscosity curves were presented in Figure 3. For the frequency sweep test (Figure 3a), W/O/W emulgels prepared by adding different primary emulsions always had higher energy storage modulus (G′) than loss modulus (G′), which indicated that these emulsions were of a gel-like structure. In Figure 3b, the viscosity of all samples decreased as the shear rate increased, and the increased addition of W/O primary emulsion resulted in a lower viscosity. Therefore, the number of primary emulsions used to formulate the double emulgels had a major impact on their stability.

Overall, these results indicated that the fresh W/O/W emulgels system formed by the addition of different W/O primary emulsions could maintain the gel-like state, but the stability of the double emulsion gradually decreased when the addition of primary emulsions exceeded 10% (*w*/*w*).

### 3.2. Effect of the SSP Concentration on the Stability of the W_1_/O/W_2_ Emulgels

Seeking to study the influence of SSPs on the stability of the double emulsions, varied concentrations of SSPs (0.5 to 2.5 wt%) were used to stabilize the W/O/W emulgels. The visual appearance pictures (Figure 4a) of the double emulgels (W_1_/O primary emulsion, 10%) showed that the gel-like appearance could be maintained after inversion. This might have led to the formation of a stable gel-like double emulsion by restricting the fluidity of the internal phase [31]. A similar phenomenon was found in the hydrogenated soybean oil double emulsions [30].

Emulsion droplet characteristics were important factors to judge the physical stability of the formed emulsion [32,33]. The particle size distribution and droplet size of the W_1_/O/W_2_ emulgels are illustrated in Figure 4c,d. The mean particle diameter decreased from 15.876 to 9.013 μm with increasing the SSP concentration from 0.5 wt% to 2.5 wt%. There was no significant (*p* > 0.05) change in double emulsion particle size between 2.0 wt% and 2.5 wt% SSPs. This effect could also be observed in particle size distribution. As the concentration of SSPs increased, the main peak around 10 μm shifted slightly to the left and decreased in intensity. These phenomena could be attributed to the sufficient number of SSPs adsorbing onto the oil–water interface to cover the larger interfacial area, resulting in the generation of smaller droplets [23]. According to earlier research, smaller oil droplets pack together more tightly, creating an emulsion system that is more stable [32]. Therefore, the stability of double emulgels was strengthened by increasing the SSPs concentration. Meanwhile, the ζ-potential values (Figure 4e) of emulsions were 40 ± 5 mV, which was slightly changed (*p* > 0.05) at different protein concentrations. The relatively high negative charge leads to strong electrostatic repulsion between the droplets to maintain good physical stability [1,34]. Additionally, the optical micrographs (Figure 4b) showed the obvious W/O/W structure features with small water droplets dispersed inside the larger oil droplets. The CLSM image (Figure 5) showed that the green on the left side of the image was the outer aqueous phase stained with FITC, and the red and black in the middle were the oil phase and inner aqueous phases. It further verifies the water-in-oil-in-water type of the W/O/W emulgels in Figure 4b.

Rheological analysis was conducted to characterize the gelling behavior and viscoelastic property of emulsions. The frequency sweeps test of double emulgels is depicted in Figure 6a. All emulsions showed a higher storage modulus (G′) than the loss modulus (G′′), and there was no crossing-over between G′ and G′′. This phenomenon could be regarded as an elastic behavior, demonstrating the establishment of a gel-like network structure [35]. This might be because SSPs have a strong gel property, which could form a viscoelastic interface layer at the oil–water interface [36]. In addition, G′ gradually enhanced as the amount of SSP concentration increased, which indicated that the stability of the W/O/W emulgels system also gradually increased. The apparent viscosity test was depicted in Figure 6b, where the viscosity of the emulsion tends to decrease as the shear rate increases from 0 to 20 1/s, indicating the shear-thinning characteristic. These results manifested that the emulsion had a pseudo-plastic behavior [37]. Moreover, the apparent viscosity of double emulgels was increased with the increase in SSP concentration. This was mainly attributed to the decrease in particle size and the increase in the specific surface area of the emulsion, which leads to an increase in the number of emulsion droplets and droplet-droplet interactions [23]. There were similar research findings by Guo et al. (2020) [32].

### 3.3. Storage Stability

To determine the potential shelf life of double emulgels, the visual appearance (W_2_: 0.5 wt%, 1.0 wt%, 1.5 wt%, 2.0 wt%, and 2.5 wt%) and optical micrographs (W_2_: 2.0 wt% and 2.5 wt%) were evaluated by observing the stable W/O/W emulgels during storage at 4 °C for 0, 10, and 30 days, respectively. In the inverted visual appearance images of Figure 7, the gel-like structure could be found in fresh double emulgels (0 days). The double emulsion sample remained gel-like at 2.5 wt% of SSP concentration after 30 days of storage. In addition, the fluidity of the formed double emulgels increased and could flow freely at values of 0.5 wt% to 1.5 wt% of the SSPs when the storage time was increased from 0 to 10 days. A similar phenomenon could be confirmed in the optical micrographs. Compared to the fresh emulsion, the droplet size of double emulgels (W_2_: 2.0 wt%, 2.5 wt%) increased significantly (*p* < 0.05) after storage. This result revealed that aggregation of the emulsions occurred during storage. This could be due to the disruption of the gel state of the double emulsions, leading to oil-off, and thus affecting the stability of the overall emulsion system [30].

### 3.4. Encapsulation of β-Carotene and Vitamin C in the W_1_/O/W_2_ Emulgels

Multiple emulsions as the special two-compartment structure could simultaneously encapsulate both water-soluble and lipid-soluble active ingredients, especially environmentally sensitive substances (ease of degradation and oxidation) [38]. The embedding stability of the W_1_/O/W_2_ emulgel system was evaluated by analyzing the encapsulation efficiency of vitamin C and β-carotene and antioxidant capacity under different storage temperatures.

#### 3.4.1. Encapsulation Efficiency

The ability of gel-like emulsions as delivery systems for the simultaneous encapsulation of hydrophilic and hydrophobic was analyzed. As shown in Figure 8, the encapsulation efficiency of vitamin C embedded in the W/O/W emulgels stabilized by 1.0% and 2.0% SSP was 87.3% and 91.2%, while the β-carotene was 99.7% and 99.8%, respectively. The encapsulation rate of β-carotene did not significantly increase (*p* < 0.05) when the concentration of SSPs in the external aqueous phase was raised from 1.0% to 2.0%, while the encapsulation rate of vitamin C did. This phenomenon further indicated that the stability of the double emulsion system increased with increasing SSP concentration. Compared with the previous study, the encapsulation efficiency of β-carotene was different to other delivery systems such as 6.0% bran wax (EE = 94.0%) [39], nanoemulsion (EE = 86.8%) [40], and zein microcapsule (EE = 91.7%) [41]. For vitamin C, the encapsulation rates of other delivery systems are as follows: whey protein-pectin (EE = 84.0%) [42], bovine serum albumin and pectin (EE = 65.3%) [43], and 2.0 mmol/g gelatin-sodium caseinate (EE = 97.0%) [44], and so forth. Therefore, the encapsulation efficiency of the double emulgel system, encapsulating vitamin C and β-carotene, was acceptable compared to other delivery systems for bioactive substances. Additionally, compared with β-carotene, there was a relatively low encapsulation efficiency of vitamin C in the W/O/W emulgels. This was mainly due to the hydrophobic environment between the inner and outer aqueous phases where more vitamin C was transferred to the outer aqueous phase during the preparation of W/O/W emulgels [9].

#### 3.4.2. Antioxidant Ability

Antioxidant capacity experiments were performed to evaluate the ability of the emulsion delivery system to protect against the functional activity materials. Figure 9 showed the retention of antioxidant capacity of vitamin C and β-carotene embedded in the W/O/W emulgel system under different temperatures. There was a similar trend in all curves at 4, 37, and 55 °C, which was that the retention of antioxidant ability decreased with increasing storage days. Specifically, as the storage time increased, the retention of ABTS+ scavenging activity of double emulgels reached 49.2–93.2%, 17.6–37.5%, and 6.0–23.5%, and the DPPH· reached 56.3–95.8%, 10.9–65.1%, and 8.8–14.7% when the temperatures were 4, 37, and 55 °C, respectively. It was obvious that vitamin C and β-carotene loaded in double emulgels had higher retention of antioxidant activity during storage compared to unencapsulated at different temperatures. On the one hand, these results showed that the reduction in vitamin C and β-carotene in double emulgels was delayed, which might be due to the existence of two interfaces that hindered oxygen exposure, slowing the degradation of the embedded substances [45]. On the other hand, the storage temperature had a greater impact on the retention of antioxidant ability of vitamin C and β-carotene encapsulated in all multiple emulsions. It illustrated that the heat treatment decreased the stability of the W/O/W emulgels.

## 4. Conclusions

In summary, the gel-like W/O/W emulsion system could be successfully formed from food-grade ingredients using a relatively rapid and simple method at different oil–water ratios. The stability of the emulsion system was influenced by different additions of W/O primary emulsion. In order to obtain a double emulgel system with better stability, the addition of primary emulsion of 10% was identified. Moreover, the gel-like structure of the fresh formation emulsion system could be engendered at the SSP concentration in the range of 0.5–2.5 wt%, and the stability of double emulgels increased with the increase in SSP concentration. Additionally, these gel-like structures of double emulsions could provide an important delivery system for simultaneous embedding hydrophilic and hydrophobic components. Co-encapsulation of vitamin C and β-carotene in this delivery system resulted in a higher encapsulation efficiency and longer shelf-life. Overall, the structure of these double emulgels provided a better appearance, longer shelf-life, and unique rheology compared to conventional W/O/W double emulsions. We are convinced that the fabrication of gel-like double emulsion using natural materials as a high encapsulation rate of bioactive molecules and stability might have great potential for applications in the food, cosmetics, and pharmaceutical industries. However, the bioaccessibility and thermal stability of hydrophobic and hydrophilic elements contained in W/O/W emulgels should be investigated in future research.

## Figures and Tables

**Figure 1 foods-11-02720-f001:**
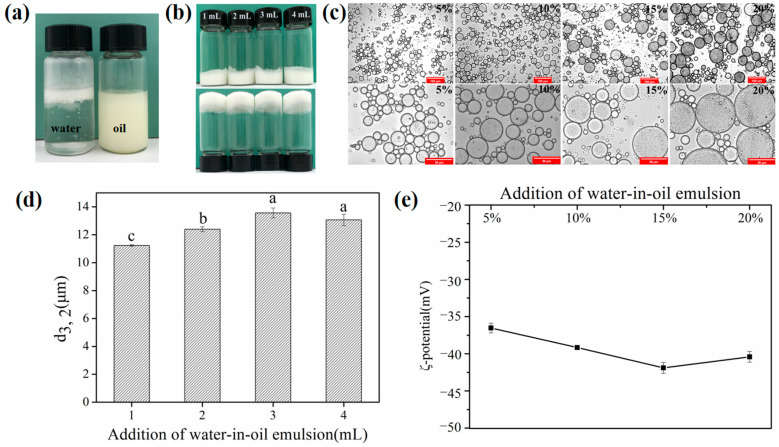
The type of primary emulsion (3.0% PGPR) (**a**); Visual appearance (**b**) and optical micrographs (**c**) of W_1_/O/W_2_ emulgels (W_2_, 1.0 wt% SSPs) prepared by different additions of W_1_/O primary emulsion (5, 10, 15 and 20%). Scale bars, 100 μm and 50 μm, respectively. Droplet size (d_3, 2_) (**d**) and ζ-potential (**e**) of W_1_/O/W_2_ emulgels with different addition of W_1_/O primary emulsion. Different letters in sub-figure (**d**) means significance difference (*p* < 0.05).

**Figure 2 foods-11-02720-f002:**
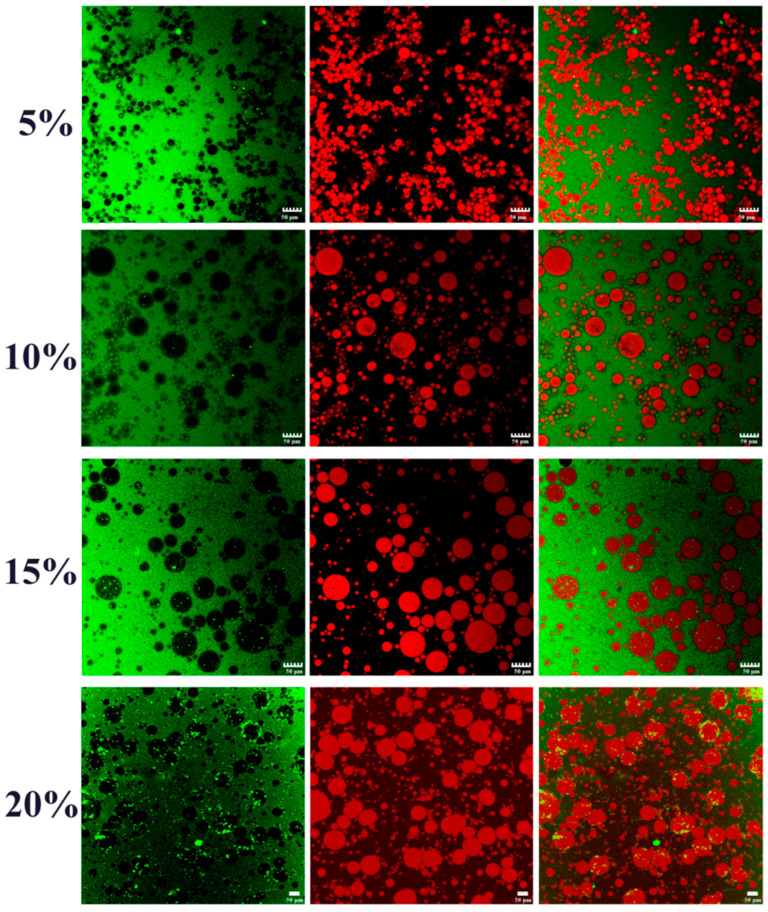
CLSM observations (dilute the sample 10 times) of the W_1_/O/W_2_ emulgels with different additions of W_1_/O emulsion (5, 10, 15, and 20%). The oil phase and outer aqueous phase (W_2_, 1.0 wt% SSPs) were dyed with the Nile red and FITC, respectively. Scale bars, 50 μm.

**Figure 3 foods-11-02720-f003:**
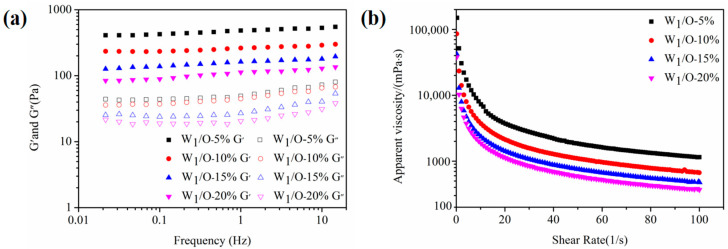
Rheological properties of W_1_/O/W_2_ emulgels (W_2_, 1.0 wt% SSPs) with different additions of W_1_/O emulsion (5, 10, 15, and 20%). Frequency sweeps curves (**a**) of the multiple emulgels at fixed strain (1.0%, with the LVR) with frequency ranging from 0.01 to 15 Hz; Apparent viscosity (**b**) of the samples with shear rate from 0 to 100 1/s.

**Figure 4 foods-11-02720-f004:**
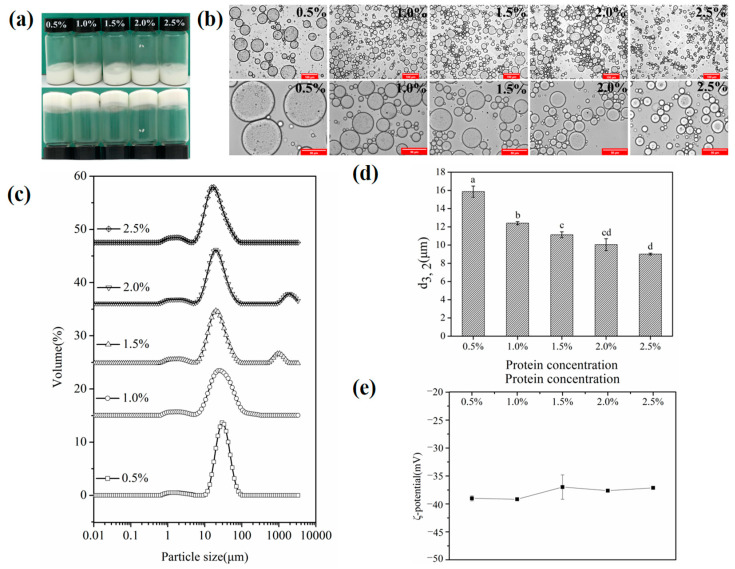
Visual appearance (**a**) and optical micrographs (**b**) (dilute the sample 10 times) (Scale bars, 100 μm and 50 μm, respectively) of W_1_/O/W_2_ emulgels (W_1_/O primary emulsion, 10%) by different concentrations of SSPs (W_2_, 0.5 wt%, 1.0 wt%, 1.5 wt%, 2.0 wt%, and 2.5 wt%). Particle size distributions (**c**), droplet size (d_3,2_) (**d**), and ζ-potential (**e**) of double emulsions with different SSP concentrations. Different letters in sub-figure (**d**) means significance difference (*p* < 0.05).

**Figure 5 foods-11-02720-f005:**
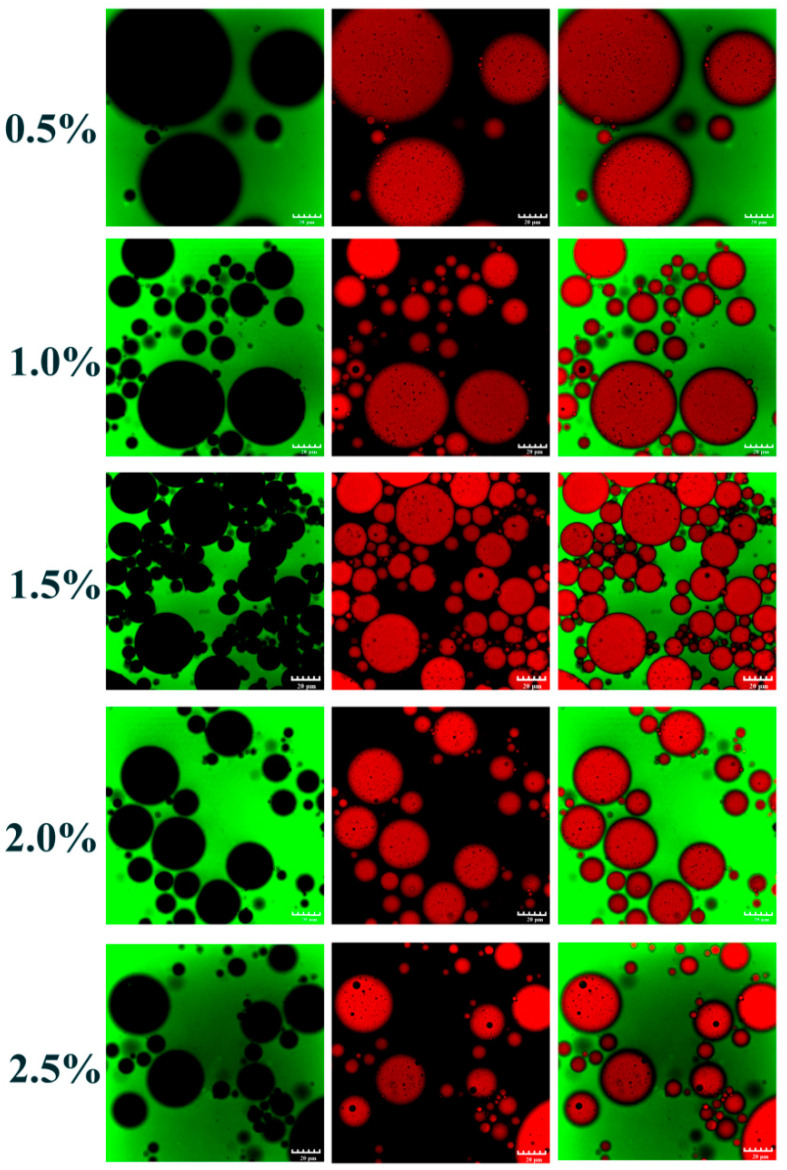
CLSM images (dilute the sample 10 times) of W_1_/O/W_2_ emulgels (W_1_/O primary emulsion, 10%) stabilized by SSP concentrations were 0.5 wt%, 1.0 wt%, 1.5 wt%, 2.0 wt% and 2.5 wt%, respectively. The oil phase and outer aqueous phase were dyed with the Nile red and FITC, respectively. Scale bars, 50 μm.

**Figure 6 foods-11-02720-f006:**
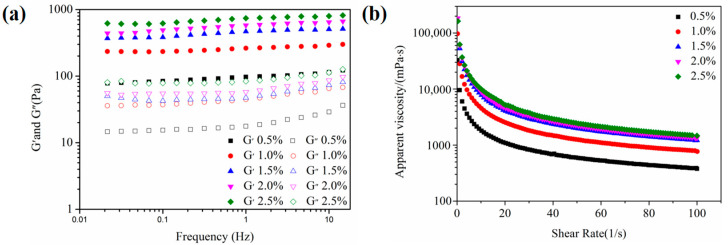
The frequency sweep curves (**a**) and apparent viscosity (**b**) of W_1_/O/W_2_ emulgels (W_1_/O primary emulsion, 10%) with different concentrations of SSPs. Scale bars, 20 μm.

**Figure 7 foods-11-02720-f007:**
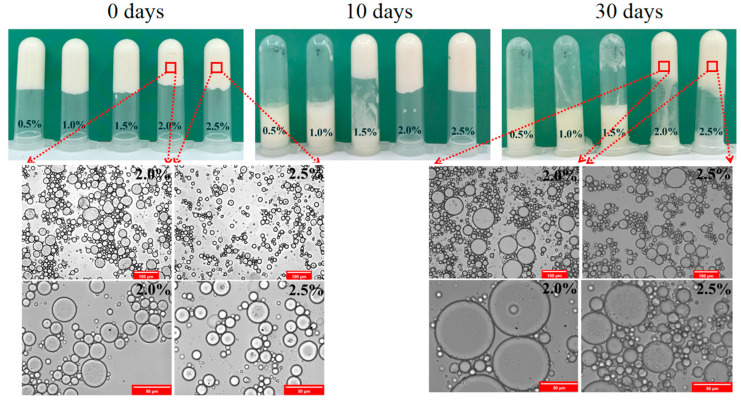
Visual appearance (W_2_: 0.5 wt%, 1.0 wt%, 1.5 wt%, 2.0 wt% and 2.5 wt%) and optical micrographs (W_2_: 2.0 wt% and 2.5 wt%) of W/O/W emulgels with different concentration of SSPs during storage at 4 °C for 0, 10, and 30 days. Scale bars, 100 μm and 50 μm, respectively.

**Figure 8 foods-11-02720-f008:**
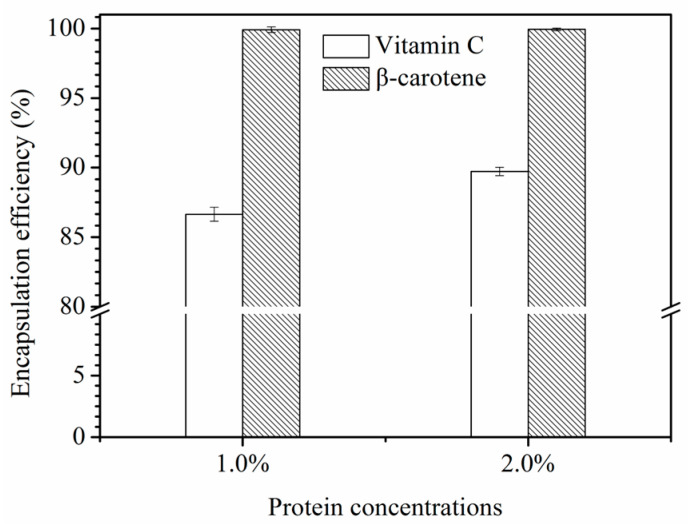
Encapsulation yield of β-carotene and vitamin C in a double emulsion.

**Figure 9 foods-11-02720-f009:**
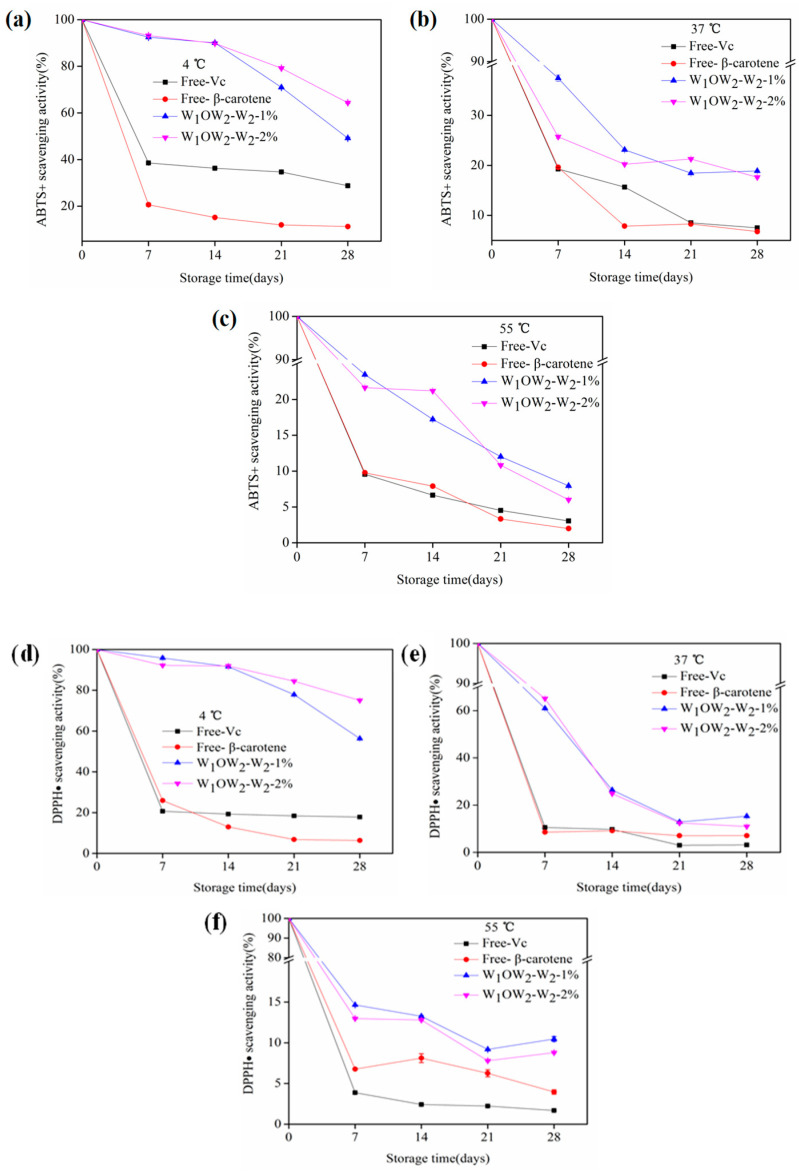
The retention of antioxidant ability of β-carotene and vitamin C in W_1_/O/W_2_ emulgels stored at 4, 37, and 55 °C for 28 days. ABTS+ (**a**–**c**) scavenging activity method. DPPH (**d**–**f**) scavenging activity method.

## Data Availability

Data are contained within the article.

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
