# Peer review of "Fabrication and Characterization of W/O/W Emulgels by Sipunculus nudus Salt-Soluble Proteins: Co-Encapsulation of Vitamin C and β-Carotene"

_foods, 2022, doi:10.3390/foods11182720_

Round 1

Reviewer 1 Report

This article describes the fabrication and characterization of W/O/W emulgels by Sipun- 2culus nudus Salt-soluble Proteins: Co-encapsulation of Vitamin 3 C and β-carotene. The article is very well written, methodology is adequate, and results are well discussed and compared with literature. I only have some minor comments and questions:

Line 92: Define PGPR at first use

Section 2.2. The time to obtain SSPs is relatively long, what are the authors thoughts on the feasibility

of this ingredient thinking on a possible scale up?

Line 149: Why were samples diluted?

Line 228: shows

Line 243: remove “e.g.”

Line 254: English check

Figure 1: Revise the title

Figure 3: Spelling check on y axis of the figure, also add “apparent”

Figure 9: Improve quality of the axis as it is hard to read.

Author Response

Response to Reviewer 1 Comments

Point 1: Line 92: Define PGPR at first use

Response 1: Thank you very much for your circumspection. PGPR has been defined at first use in the revised version. (Please see Line 90)

Point 2: Section 2.2. The time to obtain SSPs is relatively long, what are the authors thoughts on the feasibility of this ingredient thinking on a possible scale up?

Response 2: Thank you very much for your circumspection. Sipunculus nudus (S. nudus) belongs to marine worms, which are rich in numerous active ingredients such as protein, polysaccharides, and fatty acids. And the SSPs of S. nudus not only had good emulsification properties, but was also rich in amino acids. Therefore, despite the long SSPs preparation time, the results of this study could provide greater application for the production of edible and nutrient-rich double emulsion delivery systems using protein-based emulsifiers.

Point 3: Line 149: Why were samples diluted?

Response 3: Thanks for the referee’s kind advice. One of the main purpose of this study was to investigate the possibility of the formation of W/O/W emulgels at different oil-water ratios. The crux to determining the formation of emulsion was to characterise the type of emulsion. Therefore, to get a clearer view of the type of emulsion formed, the samples were diluted during the microstructural analysis (optical microscopy and confocal laser scanning microscopy (CLSM)).

Point 4: Line 228: shows

Response 4: Thank you very much for your circumspection. “Figure 1a showed” is changed to “Figure 1a showed” in the revised version. (Please see Line 229)

Point 5: Line 243: remove “e.g.”

Response 5: Thank you very much for your circumspection. “e.g.” has been removed in the revised version. (Please see Line 244)

Point 6: Line 254: English check

Response 6: Thank you very much for your circumspection. The sentence has been checked in the revised version. (Please see Line 255)

Point 7: Figure 1: Revise the title

Response 7: Thank you very much for your circumspection. The title of Figure 1 has been revised in the revised version. (Please see Line 271-275)

Point 8: Figure 3: Spelling check on y axis of the figure, also add “apparent”

Response 8: Thank you very much for your circumspection. Spelling on y axis of the Figure 3 has been modified in the revised version.

Point 9: Figure 9: Improve quality of the axis as it is hard to read.

Response 9: Thanks for the referee’s suggestion. We have modified the y axis of Figure 9, and the detailed revision could be found in the revised version.

Reviewer 2 Report

The subject is very interesting and innovative dealing with the problematics of complex emulsions that can be used for different purposes within the food and drug delivery industries. The findings of this study can be helpful for further research, which makes the results significant for the scientific peers, as well as for the food and pharmaceutical industry worldwide.

Review Comments

The objective of the article Fabrication and Characterization of W/O/W Emulgels by Sipunculus nudus Salt-soluble Proteins: Co-encapsulation of Vitamin 3 C and β-carotene is preparation of the stable W/O/W emulgels using protein isolated from marine worm species Sipunculus nudus as an emulsifier of outer aqueous phase at different oil to water ratios. The subject is very interesting and innovative dealing with the problematics of complex emulsions that can be used for different purposes within the food and drug delivery industries. The findings of this study can be helpful for further research, which makes the results significant for the scientific peers, as well as for the food and pharmaceutical industry worldwide.

In the Introduction part of the manuscript, the subject was well addressed and citation of the literature is adequate and up to date.

The Material and Methods are well presented and the methods used are appropriate for this kind of research.

The Result and the Discussion part of the article tackle the subject of the research appropriately, and elaborate the findings well, by making appropriate conclusions and references to other previously conducted researches.

The References are up to date and refer to the subject correctly.

My comments are:

1.      Use synonyms when a word is repeated within the sentence multiple times, if possible.

2.      I suggest the English language be checked for minor corrections.

Author Response

Response to Reviewer 2 Comments

Point 1: Use synonyms when a word is repeated within the sentence multiple times, if possible. 

Response 1: Thanks for the referee's kind advice. We have replaced some words that is repeated within the sentence multiple times. The detailed revisions include:

  • Replaced "form" by "develop" in line 60.
  • Replaced "form" by "maintain" in line 61
  • Replaced "form" by "create" in line 63
  • Replaced "increased" by "enhanced" in line 68, 321
  • Replaced "shown" by "illustrated" in line 295

Point 2: I suggest the English language be checked for minor corrections.

Response 2: Thanks for the referee’s suggestion. We have modified the manuscript as indicated, and the detailed revision could be found in the revised version.

Reviewer 3 Report

Referee report

Title: Fabrication and Characterization of W/O/W Emulgels by Sipunculus nudus Salt-soluble Proteins: Co-encapsulation of Vitamin C and β-carotene

Dear Editor,

The paper presents interesting results that may be useful in various branches of the food industry, but the way of their presentation should be better. This article can be recommended for publication after revision, and in my opinion, before accepting the article, the following changes (marked in color) should be made.

Abstract

Line 19 Should be Italic

Key-words

are low informative

Line 65  Dot should be after citation

Line 78-85 This fragment should be included in the chapter on measurement methodology

Line 92 PGPR please explain shortcut and degree of purity

Materials

Please provide details of purity for all components 

Line 101 Deionized water …..please add conductivity

Line 102, 105, 203 Was centrifuged at 4390 g …..please precise this unit

Line 113, 115 and others Dot after citations

Line 139 Zeta potential is calculated  on the measured electrophoretic mobility, zeta potential is not measured, because cannot be measured, it is a major factual error.

Results

Please refer in the introductory part and in the discussion the analogous results. For example from these propositions:

Zeta potential and droplet size of n-tetradecane/ethanol (protein) emulsions, B: Biointerfaces

Colloids and Surfaces 25 (2002) 55-67

Edible films made from blends of gelatin and polysaccharide-based emulsifiers - A comparative study, Food Hydrocolloids 96 (2019) 555-567

Release kinetics and antimicrobial properties of the potassium sorbate-loaded edible films made from pullulan, gelatin and their blends,  Food Hydrocolloids 101(2020) 105539

Line 140 Please precise point 2.5, Characterization is low precise

Line 174, 202 According to chemical nomenclature, n-heptane “n” should be Italic

Line 180 Correct English style

Line 192 Correct numbering

Line 195 In my opinion symbols with subscripts would be more correct.

Line 209 Please explain why time 6 min was selected?

Line 216-217, 265 Correct this sentence.

Figure 1 Standardize the signatures, the capital letter should appear in the following subsections

Figure 3 Errors in axle signatures

Line 302 Should be mV (according SI)

Line 315 It is not a chemically correct phrase, it is a colloquial expression.

Figure 6 Errors in axle signatures

Figure 8 Correct legend

Conclusions

Please emphasize in the Conclusions section how this paper contributes to new fundamental understanding for food domain.

Line 433 I do not understand this sentence???

Bibliography should be more improved

In vitro, systematic names and similar according to nomenclature should be Italic

Full names of journals or full citation

Conclusions

Please emphasize in the Conclusions section how this paper contributes to new fundamental understanding for food domain.

I can recommend this article, but after major revision.

Author Response

Response to Reviewer 3 Comments

Point 1: Abstract: Line 19. Should be Italic

Response 1: Thank you very much for your circumspection. “Sipunculus nudus” has been modified to italic in the revised version. (Please see Line 19)

Point 2: Key-words: are low informative

Response 2: Thanks for the referee’s kind advice. We have rewritten the Key-words, and the detailed revision could be found in line 29-30.

Point 3: Line 65. Dot should be after citation

Line 113, 115. and others Dot after citations

Response 3: Thank you very much for your circumspection. The format of citation has been modified in the manuscript.

Point 4: Line 78-85. This fragment should be included in the chapter on measurement methodology

Response 4: Thanks for the referee’s kind advice. We have rewritten that fragment, and the detailed revision could be found in line 78-85.

Point 5: Line 92. PGPR please explain shortcut and degree of purity

Materials: Please provide details of purity for all components

Response 5: Thank you very much for your circumspection. The details of purity for all components of Material have been provided in the revised version. (Please see Line 89-94)

Point 6: Line 101. Deionized water …please add conductivity

Response 6: Thank you very much for your circumspection. The conductivity of distilled water has been added in the revised version. (Please see Line 99)

Point 7: Line 102, 105, 203. Was centrifuged at 4390 g …..please precise this unit

Response 7: Thank you very much for your circumspection. We have precised the centrifugal conditions and units , and the detailed could be found in the revised version.

Point 8: Line 139. Zeta potential is calculated on the measured electrophoretic mobility, zeta potential is not measured, because cannot be measured, it is a major factual error.

Response 8: Thanks for the referee’s kind advice. “The ζ-potential was measured” is changed to “The ζ-potential was calculated” in the revised version. (Please see Line 138)

Point 9: Results:

Please refer in the introductory part and in the discussion the analogous results. For example from these propositions:

Zeta potential and droplet size of n-tetradecane/ethanol (protein) emulsions, B: Biointerfaces Colloids and Surfaces 25 (2002) 55-67

Edible films made from blends of gelatin and polysaccharide-based emulsifiers - A comparative study, Food Hydrocolloids 96 (2019) 555-567

Release kinetics and antimicrobial properties of the potassium sorbate-loaded edible films made from pullulan, gelatin and their blends, Food Hydrocolloids 101(2020) 105539

Response 9: Thanks for referee’s suggestion. We have improved the discussion and added some references relative with the study, and the detailed revision could be found in the manuscript.

Point 10: Line 140.

Please precise point 2.5

Characterization is low precise

Response 10: Thank you very much for your circumspection. We have revised point 2.5 in the revised version. (Please see Line 143)

Point 11: Line 174, 202.

According to chemical nomenclature

n-heptane “n should be Italic

Response 11: Thanks for the referee’s kind advice. We revised “n” of n-heptane in italic according to chemical nomenclature, and the detailed revision could be found in the revised version. (Please see Line 176, 204)

Point 12: Line 180. Correct English style

Response 12: Thank you very much for your circumspection. “with slightly modified” is changed to “with some slight modifications” in the revised version. (Please see Line 183)

Point 13: Line 192. Correct numbering

Response 13: Thank you very much for your circumspection. We have revised the format of Eqs (1), and the detailed revision could be found in line 193.

Point 14: Line 195. In my opinion symbols with subscripts would be more correct.

Response 14: Thanks for the referee’s kind advice. We have revised the symbols with subscripts in the revised version. (Please see Line 193-197)

Point 15: Line 209. Please explain why time 6 min was selected?

Response 15: Thanks for the referee’s suggestion. It was a writing mistake for “at 734 nm after 6 min”, we do carried out the lipid extraction operation at 734 nm after 30 min according relevant reference. We have revised it and the detailed revision could be found in line 211.

  • Tian, H., Xiang, D., Li, C. Tea polyphenols encapsulated in W/O/W emulsions with xanthan gum–locust bean gum mixture: Evaluation of their stability and protection. International Journal of Biological Macromolecules. 2021, 175, 40-48.

Point 16: Line 216-217, 265. Correct this sentence.

Response 16: Thank you very much for your circumspection. We have corrected the sentence in the revised version, and the detailed revisions could be found in line 218-220, 267.

Point 17: Figure 1.

Standardize the signatures, the capital letter should appear in the following subsections

Response 17: Thank you very much for your circumspection. The title of Figure 1 has been revised in the revised version. (Please see Line 271-275)

Point 18: Figure 3

Errors in axle signatures

Response 18: Thank you very much for your circumspection. The axle signatures in Figure 3 have been modified in the revised version.

Point 19: Line 302. Should be mV (according SI)

Response 19: Thank you very much for your circumspection. “mv” was changed to “mV” in the revised version. (Please see Line 306).

Point 20: Line 315. It is not a chemically correct phrase, it is a colloquial expression.

Response 20: Thanks for the referee’s suggestion. We have replaced the colloquial expression in the sentence with a suitable expression, and the detailed revision could be found in line 320.

Point 21: Figure 6.

Errors in axle signatures

Response 21: Thank you very much for your circumspection. The axle signatures in Figure 6 have been modified in the revised version.

Point 22: Figure 8

Correct legend

Response 22: Thank you very much for your circumspection. The legend in Figure 8 has been modified in the revised version.

Point 23: Conclusions

Please emphasize in the Conclusions section how this paper contributes to new fundamental understanding for food domain.

Response 23: Thanks for the referee’s kind advice. We have added some new fundamental understanding for food domain in this section. The detailed revision could be found in line 430-437.

Point 24: Line 433

I do not understand this sentence???

Response 24: Thank you very much for your circumspection. The sentence has been modified in the revised version. (Please see Line 444)

Point 25: Bibliography: should be more improved

In vitro, systematic names and similar according to nomenclature should be Italic

Full names of journals or full citation

Response 25: Thanks for the referee’s kind advice. We have revised the format of the reference, and the detailed could be found in revised version.

Round 2

Reviewer 3 Report

ok